

# Limited initial impacts of biomass harvesting on composition of wood-inhabiting fungi within residual stumps

Cédric Boué[1],*, Tonia DeBellis[2,3,*], Lisa A. Venier[4], Timothy T. Work[1] and Steven W. Kembel[1]

[1] Département des Sciences Biologiques, University of Québec at Montreal, Montréal, QC, Canada
[2] Department of Biology, Concordia University, Montreal, QC, Canada
[3] Department of Biology, Dawson College, Montreal, QC, Canada
[4] Great Lakes Forestry Centre, Canadian Forest Service, Natural Resources Canada, Sault Ste Marie, ON, Canada
* These authors contributed equally to this work.

Corresponding author
Timothy T. Work,
work.timothy@uqam.ca

## ABSTRACT

Growing pressures linked to global warming are prompting governments to put policies in place to find alternatives to fossil fuels. In this study, we compared the impact of tree-length harvesting to more intensive full-tree harvesting on the composition of fungi residing in residual stumps 5 years after harvest. In the tree-length treatment, a larger amount of residual material was left around the residual stumps in contrast to the full-tree treatment where a large amount of woody debris was removed. We collected sawdust from five randomly selected residual stumps in five blocks in each of the tree-length and full-tree treatments, yielding a total of 50 samples (25 in each treatment). We characterized the fungal operational taxonomic units (OTUs) present in each stump using high-throughput DNA sequencing of the fungal ITS region. We observed no differences in Shannon diversity between tree-length and full-tree harvesting. Likewise, we observed few differences in the composition of fungal OTUs among tree-length and full-tree samples using non-metric multidimensional scaling. Using the differential abundance analysis implemented with DESeq2, we did, however, detect several associations between specific fungal taxa and the intensity of residual biomass harvest. For example, *Peniophorella pallida* (Bres.) KH Larss. and *Tephromela* sp. were found mainly in the full-tree treatment, while *Phlebia livida* (Pers.) Bres. and *Cladophialophora chaetospira* (Grove) Crous & Arzanlou were found mainly in the tree-length treatment. While none of the 20 most abundant species in our study were identified as pathogens we did identify one conifer pathogen species *Serpula himantioides* (Fr.) P. Karst found mainly in the full-tree treatment.

# INTRODUCTION

Residual forest biomass, including non-merchantable tree-tops and branches, may serve as a renewable feedstock for bioenergy and an alternative to fossil fuels

(*Mabee & Saddler, 2010*; *De Jong et al., 2017*), and is expected to provide up to 40% of the world's energy by 2050 (*GEA, 2012*). In regions that have already begun to transition towards increased reliance on bioenergy such as Fennoscandia, the extraction of forest biomass for bioenergy can reduce volumes of residual logging material after harvest by 42–65% (*Rudolphi & Gustafsson, 2005*; *Eräjää et al., 2010*). However, increased utilization of residual biomass may pose significant conservation risks for many organisms that rely on deadwood either as a habitat or as a resource (*Walmsley & Godbold, 2009*; *Toivanen et al., 2012*). Furthermore, because of intensive harvest of timber forest products, numerous fungi are threatened with extinction (*Stokland, Siitonen & Jonsson, 2012*). In Fennoscandian boreal forests, numerous fungal species are dependent on deadwood (*Siitonen, 2001*) and more than 40% of polypore fungi are red listed (*Kotiranta et al., 2019*).

Managed forests contain a lower quantity of deadwood than natural forests (*Vallauri, Andre & Blondel, 2003*; *Debeljak, 2006*; *Fridman & Walheim, 2000*). For example, *Siitonen (2001)* has shown that in clear-cut managed boreal forests, the quantity of coarse wood debris (>10 cm diameter) (i.e., stumps, tree tops, and logs) is less than 10% of the volume observed in natural forests. Some European forests are currently subject to harvest for biofuel purposes. In these forests, residual biomass will be more intensely harvested than in clear-cut forests. For example, *Eräjää et al. (2010)* showed that the average residual biomass left on the ground in clear-cut forests is 42.3 m$^3$ ha$^{-1}$ while in forest fuel harvesting the quantity of residual biomass remaining is 26.0 m$^3$ ha$^{-1}$. Numerous studies have shown that intensive residual biomass harvest can reduce species richness and abundance of some non-saproxylic organisms (i.e., arthropods: spiders, beetles (*Work, Brais & Harvey, 2014*), oribatida (*Battigelli et al., 2004*); mammals (*Sullivan et al., 2011*)) and saproxylic organisms (i.e., arthropods: beetles: *Jonsell, 2008*; fungi: *Bader, Jansson & Jonsson, 1995*; *Stokland & Larsson, 2011*).

Residual biomass is an important resource for many organisms and for forest ecosystem function. Coarse woody debris (stumps, logs) is an important habitat and resource in forests for numerous heterotrophic and decomposer organisms (*Harmon et al., 1986*; *De Jong & Dahlberg, 2017*) including beetles (*Jonsell & Schroeder, 2014*), lichens (*Svensson et al., 2016*), bryophytes (*Caruso & Rudolphi, 2009*) and wood inhabiting-fungi (*Jonsson & Kruys, 2001*; *Nilsson, Hedin & Niklasson, 2001*; *Toivanen et al., 2012*; *Kubart et al., 2016*). Futhermore, decomposition of wood by organisms living on coarse woody material including saproxylic fungi (*Fukasawa, Osono & Takeda, 2009*) plays an important role in nutrient cycling and degradation of organic matter (*Boddy & Watkinson, 1995*; *Laiho & Prescott, 2004*). Although most studies of fungal community composition in forests have focused on coarse woody debris, fine woody debris also appears to be an important habitat for many fungi (*Kruys & Jonsson, 1999*; *Juutilainen et al., 2011*; *Küffer et al., 2008*). For example, *Nordén et al. (2004)* found that for the same volume of broadleaf tree woody debris, the diversity of *Ascomycota* is higher in fine woody debris than in coarse woody debris. Decreasing the amount of deadwood and in particular fine woody debris after intensive logging in forest fuel harvesting can also indirectly affect fungi, for example, by modifying soil microclimate and influencing fungal networks (*Ódor et al.,*

*2006*; *Brazee et al., 2014*). In Norway spruce dead wood, decreasing fungal diversity can lead to reduced rates of decomposition in the early stages (*Valentín et al., 2014*).

Stumps can provide a long-term resource for wood-decaying species compared with smaller diameter woody debris such as branches (*De Jong & Dahlberg, 2017*; *Suominen et al., 2018*), because their rate of decomposition is slower and stumps are often larger and persist longer in the forest landscape than smaller woody debris (*Holeksa, Zielonka & Żywiec, 2008*). For wood-inhabiting fungi, biomass harvesting may reduce habitat availability and connectivity (*Hanski, 2005*) thus influencing community composition (*Nordén et al., 2013*) and species occurrence (*Hanski, 1998*). Intensive harvesting of biomass lead to a decrease in the occurrence of fungal fruiting bodies on stumps and wood material >2 cm diameter 5 years after clear-cut harvesting (41% less in the forest fuel harvesting than clear cut) (*Toivanen et al., 2012*). In this context, stumps may be one of the only available substrate for fungi after harvesting.

Following clear-cutting, the increase of exposure of stump surfaces tends to promote fungal pathogen infestations (*Oliva, Thor & Stenlid, 2010*). Some pathogens, such as the basidiomycete *Heterobasidium annosum* (Fr.) Bref., can spread via mycelia from residual stumps to living tree root systems after harvesting, leading to increased disease and tree mortality. This pathogen also spreads through the air via spores that are deposited most often on tree stumps. Forest pathogen infestations can have major ecological and economic impacts, as for *Heterobasidium annosum* in the northern hemisphere (*Garbelotto & Gonthier, 2013*) and is responsible for significant economic losses in Europe (*Woodward et al., 1998*). Moreover, it has already been demonstrated in Fennoscandinavian forests, that the presence of the pathogen *Heterobasidium parviporum* Niemelä & Korhonen on the Norway spruce population can affect forest regeneration (*Piri & Korhonen, 2001*). For these reasons, it is important to monitor potential infestations caused by pathogens after forest harvesting in order to be able to detect potential economic and ecological losses.

In this study we evaluated the response of fungal communities in residual stumps 5 years post-harvest in two forest harvesting treatments with differing levels of fine and coarse woody debris for use as a biomass feedstock. In the tree-length harvesting treatment, 84 m$^3$ ha$^{-1}$ (SE = 15 m$^3$ ha$^{-1}$) of woody material was left behind after cutting, whereas 28 m$^3$ ha$^{-1}$ (SE = 3 m$^3$ ha$^{-1}$) was left after cutting for the full-tree harvesting treatment. We hypothesized that fungal community structure will change, and that fungal diversity will be lower after the full-tree forest harvesting treatment, since intensive harvesting of residual biomass around stumps would cause a decrease in the resources available for wood-inhabiting fungi and a resulting decline in fungal diversity.

## MATERIALS AND METHODS

### Study site

We sampled fungi from stumps at the Island Lake Experimental Research Site (47°42′N, 83°36′W), 30 km southwest of Chapleau, ON, Canada (Fig. 1A). The Island Lake Experimental Research site is a replicated silvicultural experiment where residual woody material including fine and coarse woody debris, cut stumps and even organic material and the upper layers of soil were removed in three increasingly intensive biomass removal
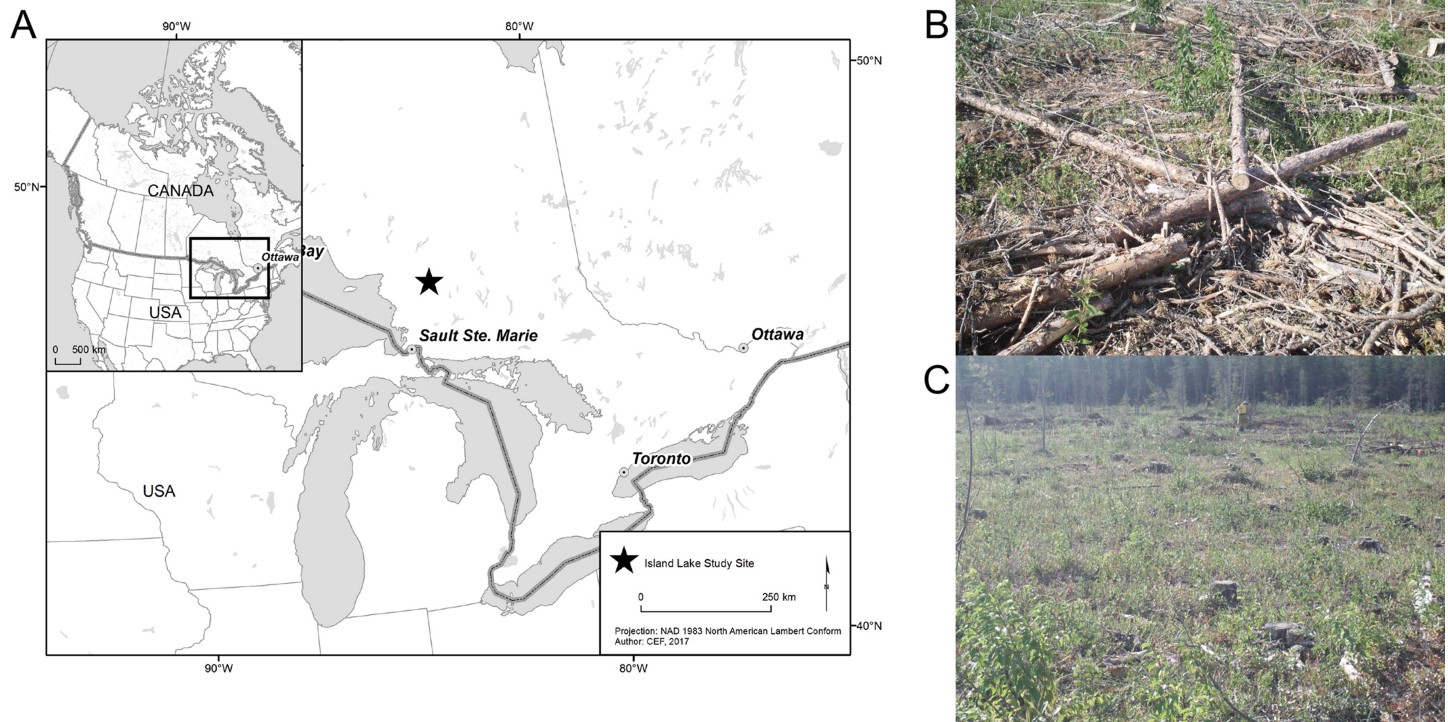

**Figure 1 (A) The study area of the Island Lake Experimental Research site in Ontario, Canada; (B) example of a tree-length harvested plot; (C) example of a full-tree biomass plot.** Photo credit: Paul Hazlett, Canadian Forest Service, Natural Resources Canada.

treatments which were applied following clear-felling. For our study, we focused on only two of these treatments; standard tree-length harvesting whereby trees were delimbed at the stump and all logging residuals were left on site (Fig. 1B) and full-tree biomass harvesting whereby all trees including non-merchantable trees were cut and removed from the site prior to being delimbed (Fig. 1C). At our sampling locations, in the tree-length treatment, an average of 84 m$^3$ ha$^{-1}$ (SE = 15 m$^3$ ha$^{-1}$) of residual biomass was left on the soil, while this volume was 28 m$^3$ ha$^{-1}$ (SE = 3 m$^3$ ha$^{-1}$) in the full-tree treatment. The tree-length treatment had 3.2–4.3 times more total deadwood biomass and 2.7–3.1 times more coarse woody material (>10 cm diameter) volume than the full-tree treatment (*Kwiaton et al., 2014*).

Because our study focused on fungal assemblages present in residual stumps, we did not study the remaining more intensive treatments where stumps were removed (stump removal and blading). All treatments were replicated five times in a randomized complete block design. The initial clear felling took place over the winter of 2010–2011 and was carried out using a Tigercat 870C feller plug and a Caterpillar 545C grapple skidder (*Kwiaton et al., 2014*). Prior to harvesting, the site was dominated by jack pines (*Pinus banksiana* Lamb.) that were planted in 1960 and had not been subjected to pre-commercial thinning. Following biomass removal, sites were replanted with both jack pine and black spruce (*Picea mariana* (Mill.) BSP) with a target stocking density of

3,300 stems/ha. The average annual temperature at this site is 1.7 °C and the average annual precipitation is 797 mm (532 mm of rain and 277 mm of snow).

## Fungal community sample collection

Fungi were collected from stumps in September 2016 (5 years post-harvest). We collected samples from five randomly selected stumps in five blocks in each of the tree-length and full-tree treatments, yielding a total of 50 samples (25 in each treatment). Fungi were collected from sawdust (ca. 100 mg) extracted from stumps using a 1 mm diameter drill. Two sawdust samples were taken from the sides of each of the stumps, including wood and bark, and pooled together. Surfaces of stumps were not surface sterilized in any way. Samples were placed directly in tubes containing a cetyltrimethylammonium bromide (CTAB) solution after collection. The drill bit was sterilized between stumps using a blowtorch for 20 s followed by a 100% bleach wash to avoid contamination. All samples were transported to the lab and stored at −20° until further processing. The use of the CTAB solution to preserve and extract fungal DNA has been recommended (*Van Burik et al., 1998*; *Kim et al., 1990*), and makes it possible to denature and eliminate contaminants of proteins (*Blin & Stafford, 1976*). We followed the CTAB extraction protocol used by *DeBellis & Widden (2006)*. The first step of the extraction consisted of breaking up sawdust pieces using a bead beater MiniBead Beadbeater-16 (BioSpec Products, Bartlesville, OK, USA) for 6 min with shaking speeds of 2,000–3,800 strokes/minute using sterilized 2.3 mm diameter stainless steel beads (BioSpec Products, Bartlesville, OK, USA).

## Fungal community sequencing

We prepared samples for high throughput sequencing by amplifying the internal transcribed spacer region using the fungal specific primer ITS1F (*Gardes & Bruns, 1993*) and ITS2 (*White et al., 1990*). The primers were designed for sequencing on the Illumina MiSeq platform by combining an Illumina sequencing adapter, a 12 nucleotide barcode to identify each sample, and the ITS1F-ITS2 primer sequences (ITS forward: 5′-CAAGCAGAAG ACGGCATACGAGATG TGACTGGAGTTCAGACG TGTGCTCTTCCGATCT xxxxxxxxxxxx CTTGGTCATTTA GAGGAAGTAA 3′, and ITS2 Reverse: 5′-AATGATACGGCG ACCACCGAGATCT ACACTCTTTCCCTAC ACGA CGCTCTTCCGATCT xxxxxxxxxxxx GCTGCGTTCT TCATCGATGC-3′. The symbol × represents the 12 barcode nucleotides used for demultiplexing of samples after sequencing). The polymerase chain reaction (PCR) reactions included 5 ul of buffer 5xHF (Thermo Scientific, Waltham, MA, USA), 0.75 M of DMSO, 0.5 M of dNTP (10 uM), 0.5 M of the reverse and forward primer, 0.25 M of polymerase *phusion* Hot start II and molecular grade water in a final volume of 25 ul.

PCR reactions were performed following initial denaturation at 98 °C for 30 s, followed by 35 cycles of 15 s at 98 °C, 30 s at 60 °C and 30 s at 72 °C, with a final elongation phase for 10 min at 72 °C. PCR products were cleaned and normalized with Invitrogen Sequalprep PCR clean-up and normalization kit. The resulting normalized samples were

pooled and sequenced on an Illumina Mi-Seq using MiSeq® Reagent Kit v3 (paired end 300 base pair) at the Université de Montréal.

Sequencing of normalized samples yielded a total of 479,034 sequences from 49 samples. We processed raw sequence data using pear (*Zhang et al., 2013*) and QIIME version 1.9.1 (*Caporaso et al., 2010*) software to assemble paired end sequences into a single continuous sequence, demultiplex sequences based on the sample they were associated with, and to eliminate low quality sequences using a quality control cutoff that eliminated all sequences with a mean quality score of 30 or less. We then binned the remaining sequences into operational taxonomic units (OTUs) using a 97% sequence similarity cutoff using the uclust algorithm (*Edgar, 2010*). We determined the taxonomic identity of each OTU using the RDP algorithm by comparison with the UNITE sequence database (*Nilsson et al., 2011*) as implemented in QIIME. We then identified the potential ecological role of the fungi detected in our samples by comparing OTUs that could be identified taxonomically to the species level with various literature sources (*Boulet, 2003*; *Kebli et al., 2012*; *Stokland & Larsson, 2011*; *Van Der Wal et al., 2017*). We removed all singleton OTUs, all samples containing less than 1,350 sequences and one outlier sample from all further analyses, leaving a total of 56,700 sequences from 42 samples after quality control and processing. We identified 1,813 OTUs from these sequences and samples. We were able to identify 625 of these OTUs to the taxonomic rank of species.

## Data analyses

In order to assure our sampling effort was sufficient to characterize the diversity of fungal communities, we calculated rarefaction curves for each sample to determine how the number of OTUs scaled with the number of sequences per sample. Rarefaction curves were based on 100 random iterations per sample.

We calculated the alpha diversity of fungal communities using the Shannon index (*Shannon & Weaver, 1949*) based on the rarefied relative abundances of OTUs. We compared Shannon diversity between treatments using linear mixed models where biomass removal treatment was treated as a fixed effect and experimental block was treated as a random effect.

We quantified the taxonomic composition of fungal communities at different taxonomic ranks in each treatment by calculating the mean abundance of OTUs identified at the rank of phylum, class, order, family, genus and species and comparing the relative abundances of each taxon between treatments using ANOVA tests on log-transformed abundance data.

To visualize the variation in fungal communities between treatments, non-metric multidimensional scaling (NMDS) analysis calculated from Hellinger transformed OTU data for communities was used, and the significance of the observed differences was determined using a permutational multivariate ANOVA (PERMANOVA; *Anderson, 2001*). The initial sample by OTU matrix rarefied to 1,350 sequences per sample was converted to a Bray–Curtis dissimilarity for both analyses.

For the PERMANOVA analysis, experimental block was included as a grouping variable for permutations. A two-dimensional NMDS failed to converge and had high stress (26.6%), and so we carried out a NMDS with three dimensions to obtain a stress of 17.8%; this three-dimensional NMDS was used for subsequent analyses. In these analyses, we used all OTUs, regardless of whether these OTUs could be resolved to recognized, named species.

In order to detect taxa or OTUs that were more abundant in the full-tree or tree-length treatment, we used the DESeq2 package (*Love, Anders & Huber, 2014*) to quantify differences in OTU and species abundances between treatments. For this analysis, the non-rarified community matrix was used but all OTUs found in three or less samples were removed from the matrix prior to the analysis. We used an adjusted *P*-value cutoff of 0.05 to consider an OTU or species as significantly differentially associated with one of the treatments.

Analyses were done using R (*R Development Core Team, 2013*) with packages picante (*Kembel et al., 2010*), vegan *(Oksanen et al., 2007)* and ggplot2 (*Wickham, 2016*), DEseq2 (*Love, Anders & Huber, 2014*), biom (*McMurdie & Paulson, 2015*), nlme (*Pinheiro et al., 2014*), multcomp (*Hothorn et al., 2016*), devtools (*Wickham & Chang, 2015*), and seqtools (*Rasmussen, 2002*).

## RESULTS

Once the community dataset was rarefied to 1,350 sequences per sample and the outlier sample was removed, a total of 42 samples and 661 OTUs remained. Rarefaction curves of the number of OTUs versus number of sequences in different samples demonstrated that our sampling effort was sufficient to quantify the diversity of the wood inhabiting fungi present in the residual stumps of the two treatments, as the observed number of OTUs within each sample reached a plateau at a lower number of sequences per sample than the number used for our analyses (1,350 sequences per sample) (Fig. S1).

The Shannon diversity of fungal OTUs per sample was marginally lower (linear mixed model; $F_{1,37} = 3.62$, *P* value = 0.06) in the full-tree treatment (1.91 ± 0.66) compared to the tree-length treatment (2.31 ± 0.59). Block as a random effect explained only 0.004% of the variance in diversity. We did not detect any statistically significant differences in the abundance of fungal taxa considering all named taxa at the ranks of phylum, class, order, family, and genus (ANOVAs of log-transformed relative abundance of each taxon between treatments for taxonomic ranks including phylum, class, order, family and genus; all *P*-values >0.05). The most abundant fungal phyla across all communities collected from stumps after residual harvesting were *Basidiomycota* (70% of sequences) and *Ascomycota* (30% of sequences).

The most abundant OTUs across both treatments included *Perenniporia subacida* (11%), an *unidentified fungus OTU* 2512 (11%) and *Scytinostorma* sp. (10%). The relative abundance of the most abundant fungal taxa in each treatment are presented in Table 1. Several taxonomically unidentified OTUs were among the most abundant OTUs (Table 1). None of the most abundant species found in our study are known to be pathogens but some could be unknown.

**Table 1 The relative abundance and taxonomic identity of the 20 most abundant fungal operational taxonomic units (OTUs) in wood from stumps after full-tree and tree-length treatments at Island Lake.** Information on the ecology, pathogenicity, and wood-decaying ability of fungi was obtained from literature sources (see Methods text for details). For ecology, pathogenicity, and wood-decaying ability, a blank cell indicates we were unable to find information for the species.

| Phylum | Species | Ecology | Pathogen | Wood decaying fungi | Relative abundance (% of sequences) | |
|---|---|---|---|---|---|---|
| | | | | | FT | TL |
| Ascomycota | Hyaloscypha sp | | no | | 0.02 | 0.02 |
| Ascomycota | Capronia leucadendri | | no | | 0.01 | 0.01 |
| Ascomycota | Chalara sp. | | | yes | 0.01 | |
| Ascomycota | Unidentified | | | | 0.01 | 0.01 |
| Ascomycota | Cenococcum geophilum | Ectomycorrhiza | | yes | 0.01 | 0.01 |
| Ascomycota | Eurotiomycetes sp. | | | yes | 0.01 | 0.01 |
| Ascomycota | Leotiomycetes sp. | | | | 0.01 | |
| Ascomycota | Helotiales sp. | | | | 0.01 | |
| Ascomycota | Leotiomycetes sp. | | | | | 0.01 |
| Basidiomycota | Perenniporia subacida | White rot | no | yes | 0.08 | 0.03 |
| Basidiomycota | Scytinostroma sp | White rot | no | yes | 0.06 | 0.04 |
| Basidiomycota | Hyphoderma praetermissum | White rot | | yes | 0.02 | 0.02 |
| Basidiomycota | Phlebia livida | White rot | no | yes | | 0.01 |
| Basidiomycota | Phlebia subserialis | White rot | no | yes | 0.01 | |
| Basidiomycota | Botryobasidium subcoronatum | White rot | no | yes | | 0.01 |
| Basidiomycota | Trichaptum fuscoviolaceum | White rot | no | yes | 0.01 | 0.01 |
| Basidiomycota | Hyphodontia floccosa | White rot | no | yes | 0.01 | |
| Basidiomycota | Phlebia tremellosa | White rot | no | yes | 0.01 | |
| Basidiomycota | Botryobasidium subcoronatum | White rot | no | yes | | 0.01 |
| Basidiomycota | Tyromyces chioneus | White rot | no | yes | | 0.01 |
| Basidiomycota | Hypholoma fasciculare | White rot | no | yes | | 0.01 |
| Basidiomycota | Hypochnicium subrigescens | | | | | 0.01 |
| Basidiomycota | Hyphoderma obtusiforme | White rot | no | yes | | 0.01 |
| Basidiomycota | Hyphoderma puberum | White rot | no | yes | | 0.01 |
| Basidiomycota | Peniophorella pallida | | | | 0.04 | |
| Basidiomycota | Botryobasidium sp. | | | | 0.01 | 0.01 |
| Basidiomycota | Dacrymyces sp. 1233 | | | | 0.01 | |
| Basidiomycota | Dacrymyces sp. 2551 | | | | 0.08 | 0.03 |
| Basidiomycota | Coniophora sp. | | | | 0.01 | |
| Basidiomycota | Unidentified | | | | | 0.02 |

There were no significant differences in fungal community composition between treatments (PERMANOVA test on Bray–Curtis dissimilarities; $r^2 = 0.04$, $P$ value = 0.2). The variation in the original dissimilarities explained by the three axes of the NMDS was 11% (Fig. S2). While there were no differences in overall community composition between treatments, some OTUs and species were differentially abundant in one of the two treatments (Table 2). For example, the species *Peniophorella pallida*, *Tephromela* sp. and

**Table 2 Fungal species that were significantly differentially abundant in full-tree or tree-length harvesting treatments at Island Lake.** Differential abundance was quantified using DeSeq2 with an adjusted *P*-value of 0.05 or lower considered as significant association with a treatment. Negative values for log-fold change in abundance between treatments corresponds to an association with the full-tree treatment, and positive log-fold changes represent an association with the tree-length treatment. Information on the ecology, pathogenicity, and wood-decaying ability of fungi was obtained from literature sources (see Methods text for details). For ecology, pathogenicity, and wood-decaying ability, a blank cell indicates we were unable to find information for the species.

| Denovo | Phylum | Species | Ecology | Pathogen | Wood decaying fungi | Log-fold change in abundance between treatments | Adjusted *P*-value |
|---|---|---|---|---|---|---|---|
| **Full-tree** | | | | | | | |
| OTU811 | *Basidiomycota* | *Peniophorella pallida* | White rot | no | yes | −5.14 | <0.01 |
| OTU484 | *Ascomycota* | *Tephromela* sp. | Lichen | no | | −3.54 | <0.01 |
| OTU1451 | *Ascomycota* | *Trichoderma citrinoviride* | soil fungus | no | | −2.71 | <0.01 |
| OTU2268 | *Ascomycota* | *Rhizoscyphus ericae* | Ectomycorrhiza | no | | −2.70 | <0.01 |
| OTU1918 | *Ascomycota* | *Scytalidium lignicola* | | no | yes | −2.47 | <0.01 |
| OTU2735 | *Ascomycota* | *Leptodontidium elatius* | Black yeast fungi | no | yes | −2.18 | <0.01 |
| OTU959 | *Ascomycota* | *Xylomelasma sp* | | | | −2.21 | <0.05 |
| OTU668 | *Ascomycota* | *Mytilinidion mytilinellum* | | no | no | −1.71 | <0.05 |
| OTU905 | *Ascomycota* | *Penicillium* sp. | | | | −1.69 | <0.05 |
| OTU875 | *Basidiomycota* | *Saitozyma* sp. | Yeast | no | yes | −1.64 | <0.05 |
| OTU1019 | *Basidiomycota* | *Serpula himantioides* | Brown rot | yes | yes | −1.56 | <0.05 |
| OTU729 | *Ascomycota* | *Rhizoscyphus ericae* | Ericoid mycorrhiza | | | −1.41 | <0.05 |
| OTU2158 | *Basidiomycota* | *Thelephora terrestris* | Ectomycorrhiza | | | −1.39 | <0.05 |
| OTU97 | *Ascomycota* | unidentified | | | | −1.35 | <0.05 |
| **Tree-length** | | | | | | | |
| OTU436 | *Basidiomycota* | *Phlebia livida* | White rot | no | yes | 3.68 | <0.01 |
| OTU747 | *Ascomycota* | Unidentified | | | | 1.81 | <0.01 |
| OTU2285 | *Ascomycota* | *Cladophialophora chaetospira* | Black yeast fungi | no | | 1.65 | <0.01 |
| OTU672 | *Basidiomycota* | *Russula* sp. | Ectomycorrhiza | | | 1.57 | <0.05 |
| OTU2105 | *Ascomycota* | *Capronia semiimmersa* | | | yes | 1.43 | <0.05 |

*Trichoderma citrinoviride* were found to be more abundant in the full-tree treatment (−5.14, −3.54, −2.71 log fold change value respectively (Table 2)). Conversely, *Phlebia lavida*, an unidentified Ascomycota and *Cladophialophora chaetospira* are examples of species found much more abundantly in the tree-length treatment (3.68, 1.81, 1.65 log-fold change values respectively (Table 2)). For the majority of the nineteen species that were found to be differentially abundant between treatments, we were unable to find any information on their ecology or pathogenicity (Table 2). The pathogen *Serpula himantioides* was differentially more abundant in the full-tree treatment.

## DISCUSSION

Contrary to our hypothesis, the diversity and composition of wood-inhabiting fungal communities in residual stumps were not significantly influenced by full-tree harvesting 5 years after harvest. It is important to stress that in both tree-length and full-tree harvest

treatments, we sampled fungal communities in residual stumps and not residual logging slash for the simple reasons smaller diameter logging residual was removed at the time of treatment. It is well documented that residual slash does serve as an important substrate for certain species (*Kruys & Jonsson, 1999*; *Juutilainen et al., 2011*; *Küffer et al., 2008*). In contrast to our findings studies from Europe that demonstrate strong effects of biomass removal on fungal composition often come from landscapes with a long history of intensive management that have yet to be created in North America (*Gu, Heikkilä & Hanski, 2002*; *Östlund, Zackrisson & Axelsson, 1997*). Forests with a longer history of logging likely contain less deadwood (*Stokland, 2001*), which for saproxylic fungi may result in habitat loss with ensuing effects related to isolation of individual habitat patches and overall reduced area of forests with old growth characteristics (*Penttilä et al., 2006*). Persistent reductions in volumes of dead wood have been shown to reduce the diversity of wood-inhabiting fungi (*Siitonen, 2001*). The minimum thresholds for harvesting biomass in Europe range between 6.1 $m^3 ha^{-1}$ (*Fridman & Walheim, 2000*) and 13 $m^3 ha^{-1}$ (*Gibb et al., 2005*) depending on harvest intensity.

Basidiomycetes and Ascomycetes dominated the fungal communities in the residual stumps in our study, a result that is, consistent with all types of fungal survey work on dead wood (*Lindner, Burdsall & Stanosz, 2006*; *Rajala et al., 2010*; *Rajala et al., 2012*). Many of the most abundant species and OTUs identified in our study, for which ecological data were available, were identified as white and brown rot fungi. These groups are characterized by their enzymatic ability to degrade wood (*Erickson, Edmonds & Peterson, 1985*) and are considered as the main decomposers of deadwood (*Stokland, Siitonen & Jonsson, 2012*). The most abundant species found in our study was *P. subacida*, a white rot fungus. This species is often reported to be present in stumps and residual branches in logged areas (*Kubart et al., 2016*; *Penttilä, Siitonen & Kuusinen, 2004*; *Brazee et al., 2012*), but it is generally not one of the most abundant species. Interestingly, despite its abundance at our study site, *P. subacida* is a vulnerable species of conservation concern in Europe (*Kubart et al., 2016*; *Parmasto, 2001*). The relative abundance of this species in our study and its scarcity in Europe are consistent with the idea that prolonged and intensive forest management has had an impact on the abundance of wood-inhabiting fungi species; in Europe, *P. subacida* is more common in forests that have limited impacts from forestry (*Penttilä et al., 2006*).

Brown rot fungi were not found to be abundant in this study, which is surprising given that brown rot fungi are more numerous in many studies on fungal diversity on the dead wood (*Rayner & Boddy, 1988*; *Rajala et al., 2012*). Brown rot fungi are considered the main decomposers of the boreal forest (*Renvall, 1995*) but this was based on the identification of fruiting bodies on conifer wood. Similarly, while ectomycorrhizal fungi can be retained by the root and stumps several years after forest harvesting and are ecologically important in boreal forests (*Hagerman et al., 1999*; *Heinonsalo, Koskiahde & Sen, 2007*), we found only one of the 20 most abundant OTUs per treatment in our study to be mycorrhizal (Table 1). None of the most abundant species in this study are known to be pathogens.

While the overall diversity and composition of fungal communities did not differ between harvesting treatments, there were individual fungal OTUs and species that were differentially abundant in one treatment. The abundance of 19 fungal OTUs differed significantly between treatments, but information about their ecology could be found only for half of these species and ecologically important fungi such as mycorrhizal fungi were found in association with both the full-tree and tree-length treatments. The pathogenic species *S. himantioides.* was more abundant on stumps in the full-tree treatment. This species is primarily a pathogen of conifers (*Seehann, 1986*) that can cause a significant loss of tree volume (*Chakravarty, 1995*) and cause root tissue death (*Seehann, 1986*). Follow-up studies will be required to determine if the association of this pathogen with the full-tree treatment will lead to increased rates of infection in this treatment.

In general, our study highlights the advantages and limitations of molecular approaches to quantify fungal community structure. Molecular analyses of the fungal taxa present in deadwood such as the present study detect a greater diversity of fungi than do studies based on the identification of fungal fruiting bodies or morphological identification of fungi in wood (*Ovaskainen et al., 2010*; *Kubartová et al., 2012*). However, molecular studies of fungal communities are currently limited by the lack of taxonomic resolution for many OTUs and the lack of information on the ecology of most of these taxa (*Rajala et al., 2012*; *Ovaskainen et al., 2010*). Alternatively, fungal community studies based on sporocarp surveys have other limitations as it will only detect species that form ephemeral, above-ground fruiting bodies (*Ovaskainen et al., 2013*), resulting with the vast majority of fungal taxa remaining unidentified (*Allmér et al., 2005*). Because very little is known on the ecology of fungi identified with environmental sequencing approaches beyond making broad categorizations such as rot fungi or mycorrhizal fungi, our ability to infer ecological differences among communities is limited even though molecular studies provide a wider range of fungal diversity. There is thus a pressing need to develop databases of the ecology of different fungal taxa that can be used to understand community and ecosystem responses to forest harvesting.

## CONCLUSIONS

Overall, our results do not suggest any difference in the diversity or structure of fungal communities between full-tree and tree-length treatments. However, some fungal species and OTUs were more abundant in one or the other of the two treatments. This study took place 5 years after harvesting in a forest dominated by a single tree species. Variation among host trees and decay stages of dead wood are important factors in the structuring of fungal communities. Future studies will be required to understand the potential long-term impacts of forest harvesting on fungal diversity.

## ACKNOWLEDGEMENTS

We would like to thank Paul Hazlett and Rob Fleming for the opportunity to conduct research at the Island Lake Biomass Harvest Experiment, Kerry Wainio-Keizer for help sampling stumps and logistics related to field work, and Melanie Desrochers for help generating the map of study sites used in Fig 1.

### Funding

This research was supported with funding by a Natural Sciences and Engineering Research Council of Canada (NSERC) Discovery grant (Kembel and Work), the Canada Research Chairs program (Kembel), the Genomics Research and Development Initiative Grant, Natural Resources Canada (Venier), and Fonds de Recherche du Québec Nature et Technologies (FRQNT) (DeBellis). The funders had no role in study design, data collection and analysis, decision to publish, or preparation of the manuscript.

### Grant Disclosures

The following grant information was disclosed by the authors:
Natural Sciences and Engineering Research Council of Canada (NSERC) Discovery grant (Kembel and Work).
Canada Research Chairs program (Kembel).
Genomics Research and Development Initiative Grant, Natural Resources Canada (Venier).
Fonds de Recherche du Québec Nature et Technologies (FRQNT) (DeBellis).

### Competing Interests

The authors declare that they have no competing interests.

### Author Contributions

- Cédric Boué conceived and designed the experiments, performed the experiments, analyzed the data, prepared figures and/or tables, authored or reviewed drafts of the paper, approved the final draft.
- Tonia DeBellis conceived and designed the experiments, performed the experiments, analyzed the data, prepared figures and/or tables, authored or reviewed drafts of the paper, approved the final draft.
- Lisa A. Venier conceived and designed the experiments, performed the experiments, contributed reagents/materials/analysis tools, authored or reviewed drafts of the paper, approved the final draft.
- Timothy T. Work conceived and designed the experiments, analyzed the data, contributed reagents/materials/analysis tools, prepared figures and/or tables, authored or reviewed drafts of the paper, approved the final draft.
- Steven W. Kembel conceived and designed the experiments, analyzed the data, contributed reagents/materials/analysis tools, prepared figures and/or tables, authored or reviewed drafts of the paper, approved the final draft.

### Data Availability

The DNA sequence data are available at the NCBI Sequence Read Archive (SRA): PRJNA557462.

## Supplemental Information

Supplemental information for this article can be found online at http://dx.doi.org/10.7717/peerj.8027#supplemental-information.

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
