# Peer review of "Limited initial impacts of biomass harvesting on composition of wood-inhabiting fungi within residual stumps"

_PeerJ, doi:10.7717/peerj.8027_

## Round 0.1 · original submission · Minor Revisions

The manuscript is well-written, with only several minor comments from reviewers. In order to improve readability of the paper it is recommended to add more visual content and shorten unnecessary descriptions. It would be a good idea, to streamline the paper by moving the details to the supplement. The reviewers have recommended adding several references. They have also made useful suggestions, and I encourage you to carefully address them.

Reviewer 1 ·

Basic reporting

The ms compares the effects of tree-length vs. full-tree harvesting on fungal diversity of the remaining stumps. I highly appreciate that although negative, the results are reported in detail. The text is well-written scientific English and easy to follow.

Introduction

An extensive number of references has been cited, and I was particularly delighted to see that the authors have found European literature (also in the Discussion part of the ms). However, despite the long and relevant list of references, some of the most recent publications seem to be missing, e.g. de Jong et al. 2017, and de Jong & Dahlberg 2017. Also, for the differences in the mycota of stumps and slash, you could check e.g. Suominen et al. 2018.

de Jong, J., Akselsson, C., Egnell, G., Löfgren, S., Olsson, B.A., 2017. Realizing the energy potential of forest biomass in Sweden - How much is environmentally sustainable?
Forest Ecology and Management 383: 3–16.
de Jong, J., Dahlberg, A., 2017. Impact on species of conservation interest of forest harvesting for bioenergy purposes. Forest Ecology and Management 383: 37–48.
Suominen, M., Junninen, K., Heikkala, O. and Kouki, J. 2018. Burning harvested sites enhances polypore diversity on stumps and slash. Forest Ecology and Management 414: 47-53.

Results

Although well-written, the Results chapter could be condensed a bit. Perhaps a bit too much emphasis is put to the taxonomic classification of the fungi. Considering the topic of the ms, ecological classification would be more relevant, because many ecological traits do not follow taxonomic classification on higher than the species level. I suggest that you delete the paragraph on lines 257-264. Also note the misclassification of Phanerochate sp. (which is a basidiomycete, not an ascomycete as you claim in table 1; see below), which may change the reported abundances. It seems clear that the great majority of the most abundant species are wood-decaying basidiomycetes.

The only figure in the ms, a map on the location of the study site, is clear and well-drawn. However, focusing the map still one step further to the block design and the selected stumps would make it more informative. Furthermore, consider adding a photo or two of the treatments, as they help readers a lot in understanding the local context. The supplementary figures are informative.

The only table is clear but it would be more informative, if there were two separate columns for abundances in each of the two treatments. Furthermore, the table contains at least one mistake. Phanerochate sp. is not an ascomycete but a wood-decaying basidiomycete in the class Polyporales. If the data has been analyzed with this misclassification, the numbers and shares of sequences reported on lines 255-264 in the text should also be checked (or rather, the paragraph on lines 257-264 could be deleted as I suggest above).

Discussion

In the Discussion I would like to see more ecological considerations and less discussion on the (molecular) methods. One perhaps the most likely reason behind the differences between the results of this study and the earlier European studies is that in this study only the stumps were sampled, whereas in European studies, the sampling usually includes also other logging residues (slash). If the main difference in between the treatments in your experiment is the presence/absence of slash, it surely affects the results whether you sample also the slash (where it is) or not. This point should be added in the Discussion. As sampling methodology was not the focus of this study, I suggest that you condense the paragraph on lines 341-368 considerably.

Minor details:

line 56 – fungi => fungal

line 57 – update the reference and citation: Rassi 2001 => Kotiranta 2019: “More than 40% of polypore fungi are redlisted.” Note that “endangered” (EN) is just one threat category among the IUCN Red List categories, the other threatened classes being critically endangered (CR), vulnerable (VU) and near-threatened (NT). (Kotiranta, H., Junninen, K., Halme, P., Kytövuori, I., von Bonsdorff, T., Niskanen, T. & Liimatainen, K. 2019. Kääväkkäät – Aphyllophoroid fungi. In: Hyvärinen, E., Juslén, A., Kemppainen, E., Uddström, A. & Liukko, U.-M. (toim.) 2019. The Red List of Finnish species 2019. Ministry of Environment & Finnish Environment Institute. Helsinki. p. 234-247.)

lines 102-103: H. annosum often spreads also vegetatively by mycelia via root contacts, although the primary mode of dispersal is via airborne spores.

lines 133-134, 146, 345: add space before the unit (m3ha-1 or mm or 2012)

lines 134-136: Change the sentence to past tense to make it clear that the values come from your experiment.

lines 248-249: delete “, although this difference was marginally significant” as this is stated at the beginning of the sentence.

line 263: format Perenniporia in italics

lines 264 and 267: Scytinostorma => Scytinostroma

lines 269-271: Add “Of” at the beginning of the sentence and change the whole sentence to past tense.

line 279: Erica => erica

line 285: What do you mean by “the other species”? Other than majority? Please clarify.

line 325: Why is Gilbertson referred here?

line 335: remove comma

line 355: Agaricles => Agaricales

lines 372-374: Delete one of the “five years after harvesting” to avoid repetition.

Experimental design

The methods are reported clearly and in detail. Unfortunately molecular methods are not my expertise.

Validity of the findings

No comment

Reviewer 2 ·

Basic reporting

no but see "general comments to the author"

Experimental design

no but see "general comments to the author"

Validity of the findings

The paper from Boué et al studies the effects two tree-harvesting methods on soil fungal communities in Ontario, Canada. The topic is interesting and timely, and answers important questions regarding the effects of forestry and removal of wood debris on soil fungal communities. The manuscript is also easy to understand and it is overall well-written. The experiment is also well-performed and replicated enough to provide solid conclusions. I am more skeptical about the hypotheses presented and the expected effects from the two treatments: for example, it is difficult for me to imagine any effects of the two treatments on fungal communities inhabiting stumps, since I would expect that communities inhabiting fine and coarse woody debris are different than the ones inhabiting the stumps. I would expect, for example, effects if most of the stumps would be removed. However, I still find these results interesting and they need to be reported.

I have some minor comments and changes but also one major comment regarding one of the analyses performed by the authors. Overall, the authors did not find any effects of treatments on the fungal communities, but they claimed to detect several associations between specific fungal taxa and the intensity of residual biomass harvest, based on DESeq2 results. Given the high diversity obtained using high-throughput DNA sequencing (often more than 2000 OTUs detected), it is very likely that some specific species respond significantly to some treatment, as it usually happens with similar analyses like indicator species analysis. Especially the species more prone to be significantly affected by some treatment are the least abundant ones, but they may be significantly affected just by chance (i.e. they are found only in two samples from the same treatment by chance). A way to avoid this and report species is to analyses only species that have high occurrences. Therefore, I encourage the authors to set a threshold for occurrences (discard for example OTUs present at least in less than 4 samples). If still there are significant OTUs but they have low abundance, I would report this at the discussion (for example just mentioning that significant OTUs had low abundance).

In L353-363: I am not sure if this is really needed. I would remove from “Studies based” to “environmental samples”, since the authors did not perform any comparison or have data to really state these sentences. Many other studies already highlighted this and in this paper there are no new findings regarding these issues.

Additional comments

I have other minor comments:
L86: what do you mean with spreading? Perhaps influencing fungal networks would be better?
L87: Wouldn’t be “reduced decomposition rates” instead of “reduced rates decomposition”?
L89: There are several papers reporting distinct communities across wood debris sizes, therefore stumps are not only a long-term resource but also may harbor distinct communities… perhaps is good to mention this as well.
L100: “Oliva” instead of “Olivia”
L106: Is there some new reference? These values were reported in 1998.
L150: It could be interesting some extra information about the sampling. For example, how many cm of sample? Did you sterilize the outer parts or removed from the analysis? This would prevent a lot of spores that may be present at the exposed part of the sample.
L168: Remove “which are the most frequently used in molecular studies of fungal communities” because these are no longer the best primers.
L180: Molarity is not provided for water. It can be mentioned “and water was added until 25 uL” or similar.
L190: Did you specify some threshold for single nucleotides or only mean quality?
L212: “…where biomass removal treatment was…”
L218: Since PERMANOVA is to statistically test the effects and NMDS is to visualize the community. I would rephrase: “A first fungal community analysis was performed using Permutational multivariate analysis of variance based on distance matrices. Matrices were calculated based on the Bray-Curtis dissimilarity index calculated from Hellinger transformed OTU data for communities rarefied to 1350 sequences per sample.
Subsequently, nonmetric multidimensional scaling (NMDS) was used to visualize community variation”
L230: Here I would not re-do the analysis but instead remove OTUs that have very frequency (i.e. <4 occurrences), perhaps it is necessary to include a column with their occurrences across the 50 samples
L236-L239: This part goes to the first paragraph of the data analysis section.
L257: change “agaricomycetes” instead of “agariomycetes”
L258-L264: Not really need to reported the sequences. I would leave only the %.
L265: Remove “ignoring identification and classification at higher taxonomic ranks”
L270: Strange to see Cenoccocum here since it is an ectomycorrhizal fungi. Perhaps it was at the part exposed..
L306-308: This is obvious, the other phylum have much lower abundant for most of the habitats. Perhaps I would remove it.
L309: how you identified the function or whether OTUs were white or brown rot fungi was not explained in M&M
L321: “not found to be abundant” or similar, instead of “were not found in large numbers”

---

## Round 0.2 · accepted · Accept

Thank you for making the suggested changes. Your manuscript is now significantly improved and acceptable for publication.